# Epigenetic Regulation in Etiology of Type 1 Diabetes Mellitus

**DOI:** 10.3390/ijms21010036

**Published:** 2019-12-19

**Authors:** Marie Cerna

**Affiliations:** Department of Medical Genetics, Third Faculty of Medicine, Charles University, Ruska 87, 100 00 Prague 10, Czech Republic; marie.cerna@lf3.cuni.cz; Tel.: +420-267102491

**Keywords:** epigenetic modifications, type 1 diabetes, HLA class II, insulin

## Abstract

Type 1 diabetes mellitus (T1DM) is caused by an autoimmune destruction of the pancreatic β-cells, a process in which autoreactive T cells play a pivotal role, and it is characterized by islet autoantibodies. Consequent hyperglycemia is requiring lifelong insulin replacement therapy. T1DM is caused by the interaction of multiple environmental and genetic factors. The integrations of environments and genes occur via epigenetic regulations of the genome, which allow adaptation of organism to changing life conditions by alternation of gene expression. T1DM has increased several-fold over the past half century. Such a short time indicates involvement of environment factors and excludes genetic changes. This review summarizes the most current knowledge of epigenetic changes in that process leading to autoimmune diabetes mellitus.

## 1. Introduction

Type 1 diabetes mellitus (T1DM) belongs to severe chronic autoimmune disorders, where mediators of autoimmune process are activated T lymphocytes, which attack pancreatic β-cells, because they recognize their autoantigens. For clinical manifestation it is necessary to destroy more than 70% tissues producing insulin. T1DM is considered to be a genetic disease with complex multifactorial heredity, since its predisposition is determined by the interaction of risk, protective and neutral alleles of approximately 50 genes together with environment. There are three main chromosomal regions that are reproducibly and statistically significantly associated with T1DM: the region of human leukocyte antigen (HLA) on chromosome 6p21, the gene for protein tyrosin-phosphatase non-receptor-type 22 (PTPN22) on chromosome 1p13, and the region of insulin (INS) gene on chromosome 11p15 [1,2].

### 1.1. HLA Class II Alleles as a Main Predisposition Genetic Factor in T1DM

HLA class II molecules are essential in the adaptive immune response. In the thymus, they participate in the selection of T cell repertoire. In the periphery, they present peptides on the surface of antigen presenting cells (APC) to the receptors of CD4+ T cells [3]. HLA class II molecules include three isotypes: HLA-DR, HLA-DQ, and HLA-DP. Each of them consists of two transmembrane chains, alpha and beta. Their extracellular parts form a peptide-binding site. HLA-DQ, particularly the alleles DQB1*02:01 and DQB1*03:02 coding for its beta chain, is the most significant predisposition molecule for T1DM. Contrary to this, HLA-DQB1*06:02 is associated with the dominant protection against T1DM. Many analyses have proved that DQB1*0302 allele is the strongest susceptibility genetic factor and that the heterozygous combination of HLA-DR4-DQA1*0301-DQB1*0302 and HLA-DR3-DQA1*0501-DQB1*0201 haplotypes results in a synergistically increased predisposition to T1DM [4].

### 1.2. Environmental Risk Factors in T1DM

Environmental risk factors are long known as a pivotal trigger of β-cell destruction (Figure 1). Early epidemiological studies have discussed viral infections as a potential cause of T1DM. The strongest candidate seems to be enteroviruses that are detected in the pancreas of T1DM patients [5].

Dietary factors involve a protective role of breastfeeding that provide passive and active immunization for infants, on one side, and on another side, a predisposition role of early exposure to cows’ milk and solid foods (fruit, root vegetables, gluten and non-gluten containing cereals, eggs). These risk factors support the hypothesis that general antigenic stimulations are more important than an actual antigen in disease process. These associations may be owing to immature immune response and insufficient tolerance in the gut [5]. Moreover, a relative deficiency of long-chain polyunsaturated fatty acids (e.g., omega-3 fatty acids), typically in many western diets, can predispose to inflammation. It should be also mentioned that toxins (chemicals or mycotoxins) in foods or water may activate autoimmune reactions [5].

A tight relationship exists between the gut microbiome and the immune system. The multiple microbiota promote the development of the immune system from the gestation period to the infancy and early childhood, particularly in the concept of induction of immune tolerance. In this context, a microbiome imbalance can cause autoreactivity in genetically susceptible individuals [6].

A lot of investigation has been made about the hygiene hypothesis. This hypothesis supposes that the increased incidence of allergic and autoimmune diseases reflects the decrease in overall infection frequency due to improved hygiene and introduction of antibiotics in industrialized countries in last century. Commensal microorganisms and parasites protect against a variety of autoimmune conditions by nonspecific allo-stimulation of innate immunity [7].

Last but not least, among environmental factors, the important point has psychological stress. Serious life events (divorce or death in the family) can activate the hypothalamic–-pituitary–adrenal (HPA) axis or the nervous system, both influence the immune cells and increase insulin resistance [8]. Furthermore, endoplasmic reticulum stress increases abnormal post-translational modification of β-cell proteins and by this way promotes generation of autoantigens.

### 1.3. Epigenetic Regulations as a Connection between Environment and Genome

Multifactorial heredity represents multiple epistasis model of inheritance, where susceptibility to certain pathological symptom is given by specific combination of multiple gene loci. Environment factors act on predisposition genes and are necessary for outburst of disease. The integrations of internal hormonal and external surrounding environments occur via epigenetic regulations of our genome, which allow adaptation of organism to changing life conditions by alternation of gene activity, by modifying gene expression. Epigenome, in fact, controls accessibility of DNA for transcription factors that regulate level of gene expression. In other words, epigenetic modifications are consequences of environment interactions with identical genotypes that result in formation of their various phenotypes [9].

Epigenetic regulations represent modifications of gene expression which do not involve nucleotide changes, but can be heritable. Main epigenetic mechanisms include DNA methylation, post-translational histone modifications, and RNA-mediated gene silencing. These epigenetic modifications are closely linked and often influence each other [10,11]. Epigenetic regulations are essential for cell differentiation, development, and protection against viruses. They are critical for the integration of endogenous and environmental signals during the life [12,13,14]. Dysregulations of epigenetic mechanisms have been associated with a number of age related disorders including cancers and autoimmunity [15,16].

A distinguishing feature of epigenetic changes in comparison with genetic changes is that they tend to be acquired in a gradual rather than an abrupt process. For example, under physiological conditions cells and tissues show a progressive loss of DNA methylation, primarily within the repeated sequences, but also in potential gene regulatory areas. In parallel, under pathological conditions cells and tissues show progressive increases in promoter methylation of selected genes, which lead to permanently gene expression silencing. These changes are highly mosaic in a given tissue and insert a high degree of epigenetic variability between cells [17]. Such epigenetic modifications could alter immune response by masking/unmasking potential antigens and by modulating immune reactions of effector cells [18].

## 2. DNA Methylation and Its Role in T1DM

DNA undergoes attachment of the methyl group on the fifth carbon of cytosine, creating a 5-methylcytosine (Figure 2), preferentially in the CpG dinucleotides. This CpG sequence is palindromic; it is present on both strands [19]. The methyl group is transferred from the S-adenosyl methionine donor (SAM) to the DNA. This transfer is carried out by DNA-methyltransferase enzymes (DNMTs).

Most CpG dinucleotides in mammalian genome are methylated. Effect of DNA methylation depends on its position in genes, but in both, in gene regulatory regions as well as in gene bodies, is important for regulation of their expression [19]. Methylation in the gene body decreases ability of RNA polymerase to transcribe a gene on the level of RNA polymerase binding, transcription initiation and elongation. For regulation of transcription initiation, the most important is methylation located in the regulatory region (in a promoter or a first intron) of the gene. Promoter methylation is generally associated with gene silencing and the mechanism of the association can be described by two non-exclusive models: (1) Methyl group directly blocks an access of transcriptional coactivators to their cognate sequences, and (2) 5-methyl-CpG is recognized by methyl-CpG-binding proteins (MBD) that induce repressive state of chromatin. In this case, it is plausible that the strength of the effect could depend on the local concentration of methylated cytosines.

Weber et al. divided gene promoters into three categories according to their C, G, and CpG density. They studied effect of promoter DNA methylation on gene expression. Promoters with low CpG density were common among tissue-specific genes, and were mostly methylated without any relationship to the gene activity. Promoters with high CpG density were more frequently found with housekeeping genes, and were usually non-methylated irrespective of the promoter activity. Promoters with intermediate CpG density were intermediary methylated, and promoter methylation correlated with gene expression. The high and intermediate DNA methylation of promoters represses gene expression [20].

However, another team did not observe a positive correlation between the promoter methylation and the gene transcription across any type of promoters [21].

According to criteria of the study above, HLA DQA1 and DQB1 gene promoters belong among low-CpG promoters, which should be methylated and their methylation should not be an obstacle to the allele expression. However, nonmethylated CpG positions are near to the beginning of transcription, where SXY boxes binding transcription factors are located. It was published that methylation in this region leads to suppression of transcription, for instance, in tumor cells [22]. A CpG rich region is located in the intron 1 of DQB1 gene, and this might be able to suppress transcription [23].

### 2.1. Monozygotic Twin Studies

T1D disease-association studies employ monozygotic (MZ) twins discordant for the disease to investigate the effect of non-genetic factors on the disease development [24]. Rakyan et al. generated genome-wide DNA methylation profiles of purified CD14+ monocytes from 15 T1DM–discordant MZ twin pairs. Monocytes are immune effector cells that give rise to tissue macrophages that have been associated with the destruction of the islet cells, causing insulin deficiency. They identified 132 T1DM–associated methylation variable positions (of them 58 were hypermethylated and 74 were hypomethylated). These included hypomethylation of HLA class II gene, *HLA-DQB1*, which carries the highest single genetic risk for T1DM (along with *HLA-DRB1*), hypomethylation of *RFXAP*, an HLA class II regulating element, hypomethylation of *NFKB1A*, an important regulator of apoptosis and inflammatory immune responses, and hypomethylation of *GAD2* which encodes GAD65, a major T1DM autoantigen involved in disease etiology. Authors also demonstrated that T1DM–associated methylation variable positions precede clinical diagnosis, and are temporally stable over many years [25].

Stefan et al. performed genome-wide DNA methylation profiles in B cell lines from 3 MZ twin pairs discordant for T1DM and 6 MZ twin pairs concordant for the disease. They identified 88 CpG sites (of them 55 were hypermethylated and 33 were hypomethylated) displaying significant methylation changes in all T1DM-discordant MZ twin pairs, including hypermethylation of *HLA-DOB* and *HLA-DQA2* genes in patients [26].

Elboudwarej et al. provided genome-wide DNA methylation profiles in peripheral blood from 7 T1D–discordant MZ twin pairs. Strong evidence for global hypomethylation of CpG sites within promoter regions in MZ twins with TIDM compared to twins without T1DM was observed [27].

There were presented a large epigenome-wide association study across 406,365 CpGs in 52 MZ twin pairs discordant for T1DM in three immune effector cell types, CD4+ T cells, CD19+ B cells and CD14+CD16- monocytes. Authors observed a substantial enrichment of differentially variable CpG positions in T1DM twins when compared with their healthy co-twins and when compared with healthy, unrelated individuals. These T1DM-associated differentially variable CpG positions were found to be temporally stable and enriched at gene regulatory elements. Evidence from cord blood of newborns who progress to overt T1DM suggested that the differentially variable CpG positions likely emerged after birth. Integration with cell type-specific gene regulatory circuits highlighted pathways involved in immune cell metabolism and the cell cycle (particularly in CD19+ B cells, there were found transcriptional regulators such as NRF1 and FOXP1 and pathways such as mTOR signaling). Consequently, authors overlapped these T1DM-associated differentially variable CpG positions with 59 T1DM genetic susceptibility loci retrieved from T1DM base, and they did not find a statistically significant enrichment of differentially variable CpG positions at these loci. This analysis provided further evidence that T1DM-associated genetic and epigenetic variants have appeared to act independently [28].

The association between DNA methylation and T1DM is supported by observation that methylation status of CD14+ monocytes and CD4+ T cells of a prediabetic quadruplet was intermediate between its affected and healthy siblings, suggesting a relationship between disease severity and DNA methylation [29].

The comparison of DNA methylation of the HLA-DQA1 gene between HLA-matched T1DM patients and healthy unrelated controls in our laboratory revealed no difference in DNA methylation of the proximal promoter of this gene. However, for the first time, the complete methylation profile of the HLA-DQA1 promoter was gained with the most methylated allele DQA1*02:01 and the least methylated DQA1*05:01 in both studied groups [30,31].

### 2.2. The Decrease of Immune Tolerance is Regulated by DNA Methylation

Many studies have implicated defects of immunological tolerance in the onset and progression of autoimmune disease, such as T1DM. Well-known immunoregulators that can suppress the proliferation of effector cells are regulatory T cells (Tregs). Tregs are a unique population of CD4+, CD25+ T cells that express the “forkhead box P3” transcription factor (FOXP3).

Epidemiological studies suggest that latent autoimmune diabetes in adults (LADA) may account for 2–12% of all cases of diabetes. The presence of autoantibodies along with islet-reactive T cells in LADA provides strong evidence that the disease process is autoimmune. LADA is thought to be a subgroup of type 1 diabetes, which has a slow procession of autoimmune destruction of β-cells. It was observed that genomic DNA methylation in CD4+ T cells from LADA patients was significantly increased compared to controls, and the *FOXP3* promoter region was hypermethylated in CD4+ T cells from LADA patients compared with controls. Subsequently, it was proved at the level of mRNA that *FOXP3* expression was decreased in diabetic patients [32].

### 2.3. Insulin Gene and Its Epigenetic Modifications

The insulin (*INS*) region is the second most important locus associated with T1DM. Many studies have been consistently confirmed that T1DM is associated with the A/T single nucleotide polymorphism (SNP) called rs689 and located in the intron 1, at position +215 bp distal to the transcriptional start site of the *INS* gene. Proximal to rs689 and in complete linkage disequilibrium with it, there is located the variable number tandem repeat (VNTR) polymorphism, which is divided into three classes. In Caucasians, SNP allele A is associated with short class I VNTR alleles, more frequent in T1DM, while SNP allele T is associated with long class III VNTR alleles. Studies of *INS* gene expression suggested that while class I alleles are associated with increased *INS* expression in the human pancreas versus class III alleles, the opposite observation has been obtained in the thymic tissue. Decreased *INS* expression of class I alleles in the thymus could lead to worse negative selection of autoreactive T cells, and then higher predisposition to autoimmunity. The variation of DNA methylation within the *INS* gene promoter is suspected to regulate *INS* gene transcription in the pancreatic β-cells and the medullary thymic epithelial cells, the two tissues that express this gene and are central to the mechanisms of T1DM [33].

The study of DNA methylation pattern of the 7 CpGs in the *INS* gene promoter revealed that T1DM patients have a lower methylation at CpGs −19, −135, and −234 (*p* = 2.10^−16^) and a higher methylation at CpG −180 than controls, while methylation was comparable at CpGs −69, −102, −206. The magnitude of the hypomethylation relative to a control population was 8–15% of the corresponding levels in controls [34].

One study highlighted the cross talk between immune responses and β-cell specific DNA methylation changes at *Ins1* and *Ins2* in islets from non-obese diabetic mice (NOD) mice, and in human β-cells in vitro. In the NOD mouse model of T1DM, inflammatory cytokines including TNF, IFNγ, IL6 and IL1B increase with age. Authors showed reduced insulin gene expression and increased percent DNA methylation at exon 2 of *Ins1* and exon 1 of *Ins2* genes in sorted β-cells from 4 week-old NOD mice cultured in media with cytokines. Moreover, increased cytokines induced mRNA expression levels of DNMTs in sorted β-cells of cultured islets from NOD mice and from human non-diabetic donors. This study suggests that increased cytokine levels associated with T1DM induce increased DNA methylation and decreased insulin mRNA levels in islets [35].

There was interest in quantifying the amount unmethylated preproinsulin DNA in the circulation as a biomarker of β-cells death. The β-cells have a much higher frequency of unmethylated CpG sites within the preproinsulin gene than other cells, and upon β-cell death these DNA sequences are released into the circulation [36]. Studies were found increased levels of unmethylated preproinsulin DNA in peripheral blood samples of patients with new-onset T1DM compared with controls [37,38]. However, although this hypothesis has been disputed, circulating demethylated amylin DNA has seemed to be a valid biomarker for β-cell death in T1DM [39].

### 2.4. Interleukin 2 Receptor α-Chain Gene and Its Epigenetic Modifications

Interleukin 2 receptor α-chain (*IL2RA*), or CD25 molecule, is part of the high-affinity IL-2 receptor complex. *IL2RA* is expressed constitutively on regulatory T cells, a population of T cells that have a potent ability to suppress autoreactive T cells, whereas is induced in other T cells. *IL-2RA* polymorphisms are associated with T1DM and other autoimmune diseases such as multiple sclerosis or rheumatoid arthritis.

The study of DNA methylation pattern of the 6 CpGs in the *IL2RA* gene promoter revealed that T1DM patients have a higher level of methylation at CpGs −373 and −456 than controls (*p* = 1.10^−4^ and *p* = 2.10^−6^ respectively). Moreover, among SNPs located in the neighboring region, it was found that twenty-eight SNPs were associated with DNA methylation at CpG −373, and sixteen of these SNPs were known to be associated with T1DM. These findings suggest that the effect of *IL2RA* risk alleles on T1DM may be partially mediated through epigenetic changes [40].

### 2.5. The Intestinal Microbiome and Epigenome in T1DM

The fetus gut is sterile. Its colonization of microbes starts at birth, and includes maternal microbiota of genital tract and colon. In neonates, the species of intestinal microbiota are not too much divergent, but they enlarge and diverge in the next two years, and remains constant through all life [41].

It was observed that the gut microbiome of individuals, before or after manifestation of diabetes mellitus, is different from that of healthy individuals. The intestinal microbiota in patients with preclinical T1DM is characterized by Bacteroidetes phylum, a lower quantity of butyrate-producing bacteria, reduced bacterial diversity, and community instability. These changes emerge after the positivity to autoantibodies that are predictive for T1DM. The gut microbiota could be involved in the progression from asymptomatic autoimmunity to its clinical manifestation rather than in the initiation of β-cell destruction [42].

Butyrate is one of the short-chain fatty acids (others are acetate and propionate) produced by bacterial fermentation in the gut. Butyrate is known to play an important role in maintaining the integrity of the epithelial layer. There is also increasing evidence that butyrate has epigenetic effects that may be very important in T1DM, where their deficiency leads to disease manifestation. Butyrate induces the methylation of promoter regions, which causes both up- and downregulation in different sets of human genes. Histone acetylation also appears to be regulated by butyrate production. Butyrate reduces lipopolysaccharide-induced inflammation in the intestine through modulation of antioxidant defense systems, nitric oxide production, and expression of inflammatory cytokines [43]. See the summary about DNA methylation in Table 1.

## 3. Histone Modifications and Their Role in T1DM

Histones undergo post-translational modifications on the specific amino acid residues in the N-terminal part of the histone. The significance of modifications is determined by the type of modification (acetylation, methylation, phosphorylation, ubiquitination, and more), the position in the nucleosome (modified residue on the histone) and the degree of modification (e.g., mono-, di-, and trimethylation). The certain modification is specifically recognized by chromatin-remodeling proteins that subsequently change the level of chromatin condensation. The resulting effect on the level of gene expression depends on the region of modifications (gene promoter, gene body, enhancer, and CpG islets), the combination of modifications and the pattern of modifications [17].

Histone acetylation, particularly histones H3 and H4, is an important marker of transcription activation, because it brings a negative charge and so keeps away the negatively charged DNA molecule. Lower chromatin condensation forms a space for the transcription complex. Acetylation is reversible, and it is performed on lysine residue. All process is controlled by histone-acetyltransferases (HATs) and histone-deacetylases (HDACs), which function as the transcription co-activators, HAT, and the transcription corepressors, HDAC, respectively. Acetylation of histones is a result of the balance between the activity of HAT and HDAC. Removal of the acetyl group causes chromatin condensation and transcription repression. A low level of acetylation can induce further histone modifications and DNA methylation. A high level of acetylation protects DNA from methylation, and reversely, DNA methylation prevents histone acetylation [9].

Histone methylation is performed on lysine or arginine residues, and includes the addition of one to three methyl groups. Histone methylation is associated with both, transcription activation or repression, depending on the gene region and extent level of modification. Monomethylation or trimethylation of lysine 4 of the histone 3 (H3K4me1 or H3K4me3) is associated with active promoters. On other side, trimethylation of lysine 9 and 27 of the histone 3 (H3K9me3 and H3K27me3) is associated with gene silencing and compacted chromatin. There is closed relationship between the methylation status of lysine 4 and lysine 9 on the histone H3 (H3K4 and H3K9) and DNA methylation. The DNA methylation is linked to the absence of H3K4 methylation and the presence of H3K9 methylation [10].

The main linkage of histone methylation with DNA methylation are DNMT proteins (Figure 3) that specifically interact with the N-terminus of H3, but only if it is unmethylated. Thus, they act as a H3K4 methylation sensor that and in the absence of H3 methylation induces de novo methylation of DNA. The link of H3K9 methylation with DNA methylation is mediated by UHRF protein which binds DNMTs and brings them to DNA [44].

The significant link between histone modifications and HLA class II expression is presented by a transcriptional coactivator Class II transactivator (CIITA). This regulatory protein not only associates with HATs, but also acts as one itself. The importance of its role in epigenetic regulation is underlined by the fact that HLA II genes can be induced by HDAC inhibitor Trichostatin A (TSA) even in the absence of CIITA [45].

### 3.1. The Studies of Natarajan’s Group

Histone modifications in T1DM were explored by a set of works by Miao. The authors observed differences in various histone modifications (H3K9Ac, H4K16Ac, H3K4me3, H3K9me2,3, H3K27me3) of genes involved in diabetes pathways between T1DM and healthy controls [46,47]. The loci of variant chromatin modifications were also located close to DQB1 and DRB1 genes: Monocytes from T1DM patients had lower levels of H3K9Ac 4 kb upstream of HLA-DRB1 and higher levels of H3K9Ac 4 kb upstream of HLA-DQB1 [47]. The increased acetylation at these sites correlates with increased transcription in the monocyte cell line [47]. However, it is not possible to decide whether the differences are a cause of the disease or result of the disease-associated hyperglycemia [48,49]. In addition, authors do not state whether the patients and controls were HLA-matched or not, so we cannot exclude the option that the observations are result of interallelic variation rather than a disease.

### 3.2. The Role of Innate Immunity

The study of inflammatory mediators, such as COX-2, in monocytes found that acetylated histone H4 expression was increased in T1DM patients compared with control subjects. COX-2 levels did not seem to follow the histone acetylation pattern, indicating that its induction may not be related to the hyperacetylation. When the diabetic group was divided into two groups on the basis of pre-diagnosed vascular complications, the histone hyperacetylation was restricted to the complication-free group, indicating that it is not associated with diabetic complications [50].

### 3.3. The Decrease of Immune Tolerance is Regulated by Histone Acetylation

Similarly to the publication on DNA methylation in CD4+ T cells from LADA patients, another study was explored whether the histone acetylation of CD4+ T cells is involved in the pathogenesis and development of LADA. In accordance with the observations of DNA methylation study, authors found the reduced global H3 acetylation in CD4+ T cells from LADA patients. The reduced H3 acetylation lever was associated with the positivity to GAD autoantibodies; the most important autoimmune marker of LADA. The expression of acetyltransferase CREBBP in LADA patients was downregulated and the expression of histone deacetylases HDAC1 and HDAC7 was upregulated. They concluded that changes in H3 acetylation in CD4+ T cells possibly contributed to the pathogenesis of LADA [51]. Unfortunately, authors did not measure the FOXP3 expression and promoter methylation status, so there is missing information about the proportion of Tregs. See the summary about histone modifications in Table 2.

## 4. RNA Interference and Their Role in T1DM

RNA interference denotes sequence-specific mRNA degradation induced by long double stranded RNA. It is an ancient eukaryotic defense mechanism against viruses and mobile elements. In mammals, endogenous RNA interference was outstripped during evolution by the current innate and acquired immunity, but its apparatus, which remains essentially intact, serves mostly the silencing pathway, which regulates endogenous gene expression. There are three well-defined RNA silencing pathways: microRNA (miRNA), small interfering RNA (siRNA), and piRNA pathways [52].

In somatic cells, miRNAs are the most abundant and functionally dominant small RNA class. During miRNA biogenesis (Figure 4), RNase III Dicer cleaves small hairpin precursors (pre-miRNAs) and produces 21–23 nucleotides long miRNAs loaded on the RNA-induced silencing complex (RISC). The siRNA pathway shares protein components with the miRNA pathway; siRNAs contain also ~22 nucleotides, and it is produced by Dicer from long double stranded RNA. The experimental gene knock-down in mammalian cells relies on short RNAs—synthetic siRNAs or expressed miRNA.

The piRNA pathway operates in the germline. Substrates for the piRNA pathway are sense and antisense transcripts from discrete genomic loci (piRNA clusters), which are produced by a complex, Dicer-independent mechanism. The piRNAs are longer than siRNAs or miRNAs (24–30 nucleotides).

The key component of RISC is an AGO protein from the Argonaute protein family (piRNAs are loaded onto Argonaute proteins from the PIWI subfamily). Mammals have four AGO proteins (AGO1–4). All AGO proteins bind miRNAs. AGO1, AGO3, and AGO4 induce translational repression. Only AGO2 is capable of endonucleolytic cleavage of cognate RNAs, which is the hallmark of siRNAs. Some miRNAs loaded on AGO2 can induce endonucleolytic cleavage upon perfect base-pairing with targets. However, a typical miRNA binding is imperfect and results in translational repression. By this way, miRNAs function as gene-specific inhibitors where miRNA networks provide a combinatorial system of post-transcriptional control of gene expression [53].

### 4.1. The Decrease of Immune Tolerance is Regulated by miRNAs

Genome-wide miRNA expression profiles of Tregs in T1DM patients described, in comparison with healthies, a significantly increased level of miR-510 and decreased levels of miR-342 and miR-191. Moreover, the miRNA comparison between Tregs and T cells found a significant higher level of miR-146a and lower level of eight specific miRNAs (20b, 31, 99a, 100, 125b, 151, 335, and 365) in Tregs, supporting their involvement in T1DM [54].

Another group analyzed the hypothesis that the failure to activate apoptosis produces uncontrolled expansion of autoreactive CD8+ T cells in diabetic patients. They compared transcriptome and corresponding miRNA expression with fate of autoreactive T cells from healthy and T1DM individuals after their exposure to islet-autoantigen. Transcriptome analysis described reduced expression of TRAIL, TRAIL-R2, FAS, and FASLG (members of the extrinsic apoptosis pathway) in T cells derived from patients, compared with T cells derived from healthies. This finding was associated with increased expression of miRNAs that are predicted to regulate these genes, particularly miR-98, miR-23b, and miR-590-5p [55].

### 4.2. The Detection of miRNAs in Peripheral Blood Mononuclear Cells

Sebastiani et al. found that miR-326 was significantly increased in peripheral blood lymphocytes from patients with T1DM and the elevated levels correlated with disease severity. Furthermore, specific targets of miR-326 (vitamin D receptor, VDR, and erythroblastosis virus E26 oncogene homolog 1, Ets1) are important immune regulators, potentially identifying pathways by which this miRNA may exert important stimulatory effects toward the development of T1DM [56]. Similar results had been observed in multiple sclerosis, in which miR-326 regulated Th-17 differentiation, and its levels were highly correlated with disease severity [16].

Salas-Pérez et al. showed that miR-21a and miR-93 are downregulated in peripheral blood mononuclear cells of T1DM patients [57]. The gene encoding the primary miR-21 (the primary transcript containing miR-21) is located within the intron region of the *TMEM49* gene. Unlike other miRNAs, the function of miR-21 has been clarified to a large extent, its over-expression patterns in cancer have generally been well established, and many of its bioinformatically predicted targets have been confirmed. Many publications have reported that miR-21 promotes Th17 cell differentiation, which mediates the development of multiple autoimmune diseases [58].

Yang et al. identified 26 miRNAs and 1218 genes differently expressed in peripheral blood mononuclear cells from newly diagnosed T1DM patients. One of the most downregulated microRNAs in T1DM was miR-146, and its expression level inversely correlated with the serum titers of GAD antibodies [59].

### 4.3. The Experimental Studies in Cultured Cells and Animal Models

Many experimental studies in cultured cells and animal models of T1DM have provided convincing evidence that miRNAs can participate in controlling β-cell fate, autoimmune damage of β-cells, and regulation of insulin synthesis and secretion [60]. In mice, the disruption of miR-155 promoted the onset of T1DM and a reduction of Treg cell number [61].

In NOD mice, Ruan et al. described a unique regulatory pathway of β-cell death that comprises miR-21, its target programmed cell death protein 4 (PDCD4), and its upstream transcriptional activator nuclear factor-κB (NF-κB). In pancreatic β-cells, c-Rel and p65 of the NF-κB family activated the *mir-21* gene promoter and increased miR-21 RNA levels; miR-21 in turn decreased the level of PDCD4, which is able to induce cell death through the Bax family of apoptotic proteins. Consequently, PDCD4 deficiency in pancreatic β-cells renders them resistant to death. Thus, the NF-κB−microRNA-21−PDCD4 axis plays a crucial role in T1DM and represents a unique therapeutic target for treating the disease [62]. See the summary about RNA interference in Table 3.

## 5. Conclusions

Epigenetic modifications influence pathogenesis of T1DM [63]. A better understanding of epigenetic mechanisms is necessary for identification of the target epigenetic pathways involved in the ethiopathogenesis of T1DM. Knowledge of the epigenetic changes in T1DM can help us to find potential biomarkers for prevention, diagnosis, prognosis, and personalized treatment of the disease [64]. Lifesaving insulin therapy unfortunately does not restore the loss of pancreatic function. Epigenetic drugs may partly prevent from the destruction of β-cells.

## Figures and Tables

**Figure 1 ijms-21-00036-f001:**
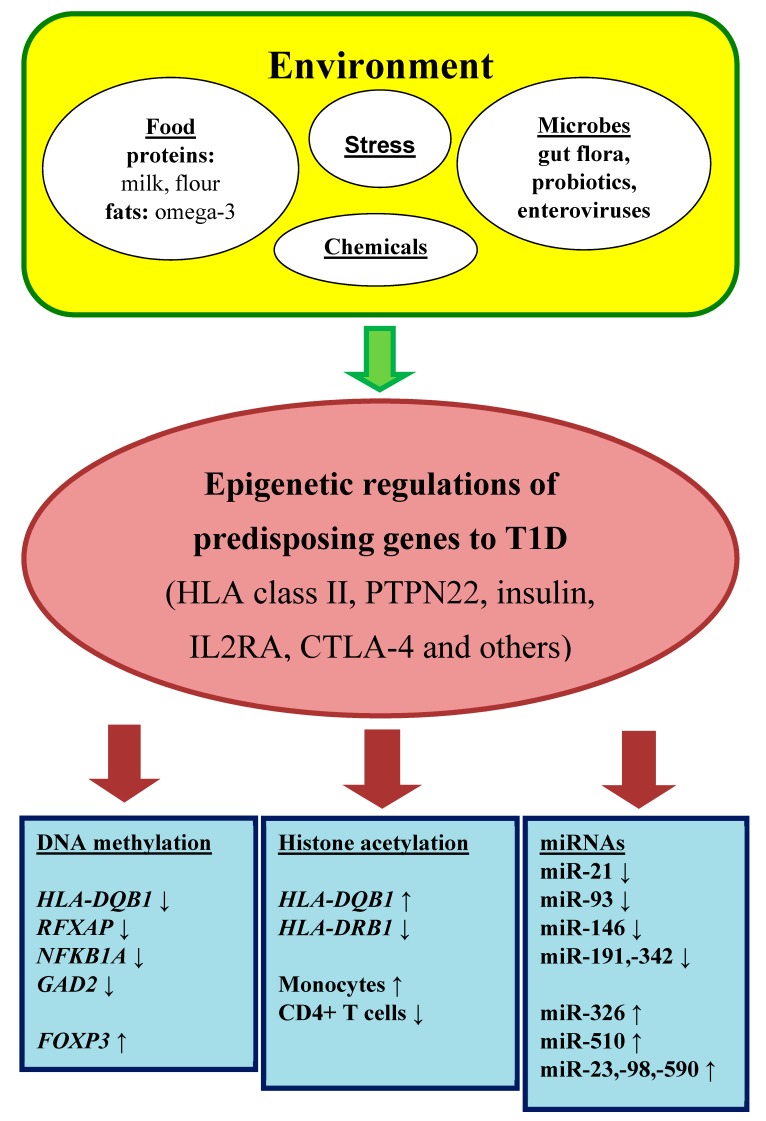
Epigenetic regulation in autoimmune diabetes mellitus.

**Figure 2 ijms-21-00036-f002:**
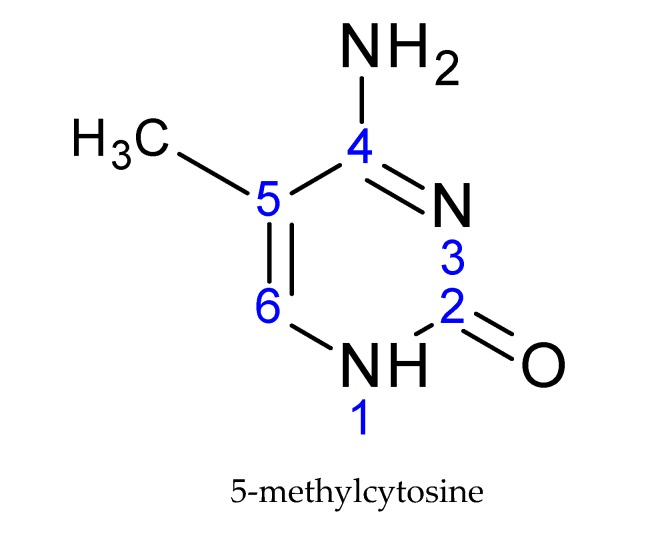
The basic structural element of DNA modification: 5-methylcytosine.

**Figure 3 ijms-21-00036-f003:**
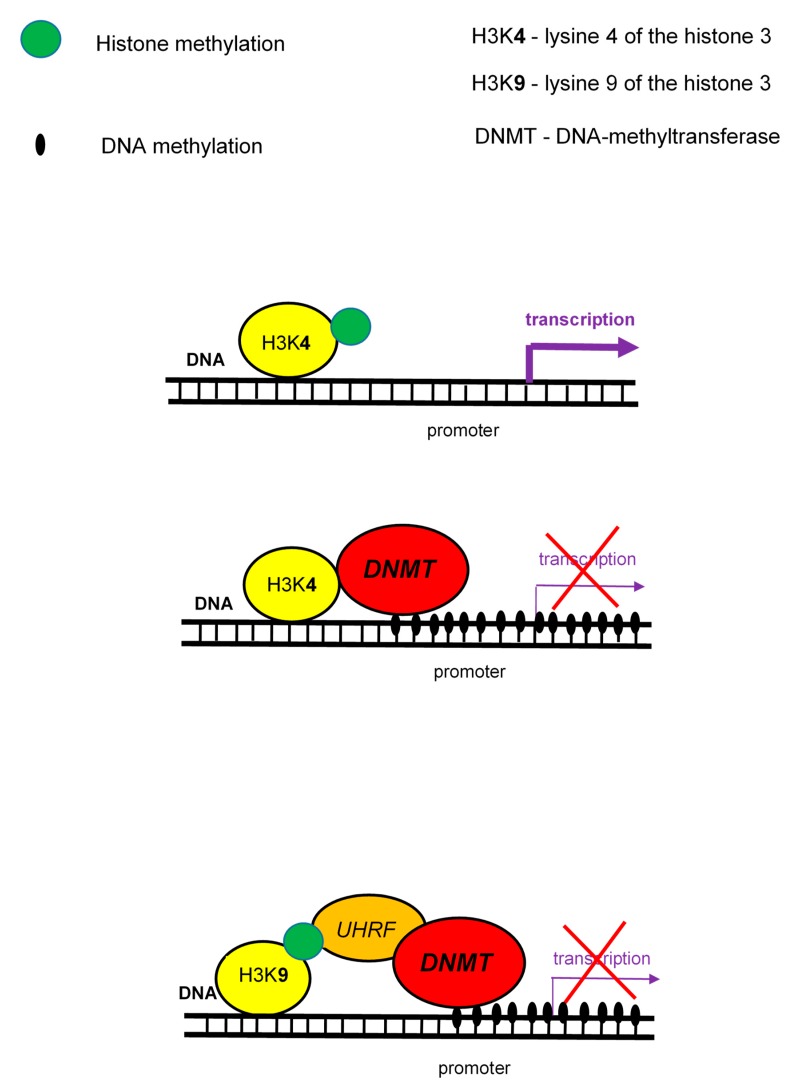
Interactions between histone methylation and DNA methylation.

**Figure 4 ijms-21-00036-f004:**
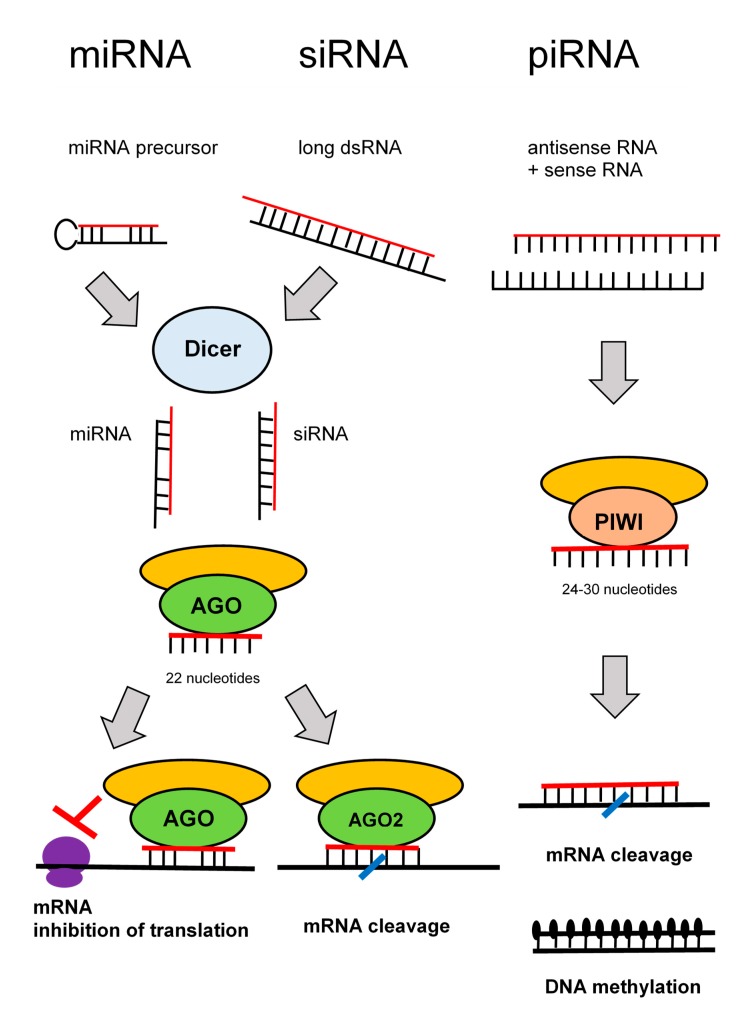
Pathways of RNA interference.

**Table 1 ijms-21-00036-t001:** The epigenetic changes found in patients with T1DM: DNA methylation.

Publication	Specific Target	Type of Cells	Results
(1st Author + Year)			
Rakyan, 2011 [25]	genome-wide profile	CD14+ monocytes	↓ *HLA-DQB1*, *RFXAP*
(MZ twins)			*NFKB1*, *AGAD2*
Stefan, 2014 [26]	genome-wide profile	B cell lines	↑ *HLA-DOB + HLA-DQA2*
(MZ twins)			
Elboudwarej, 2016 [27]	genome-wide profile	peripheral blood	global hypomethylation
(MZ twins)			
Paul, 2016 [28]	genome-wide profile	CD4+ T cells	long time stable variabilities
(MZ twins)		CD19+ B cells	in regulatory regions
		CD14+ monocytes	
Disanto, 2013 [29]	genome-wide profile	CD14+ monocytes	association with severity
(MZ quadruplet)		CD4+ T cells	of disease
Čepek, 2016 [31]	*HLA-DQA1* gene	peripheral blood	no differences between
		CD14+ monocytes	patients versus healthies
Li, 2011 [32]	genome-wide profile	CD4+ T cells	↑ *FOXP3*
Fradin, 2012 [34]	*INS* gene promoter	leucocytes	4 CpG variabilities (3↓ + 1↑)
Belot, 2013 [40]	*IL2RA* gene promoter	peripheral blood	2 CpG variabilities (both ↑)

↓ Decrease of DNA methylation (hypomethylation). ↑ Increase of DNA methylation (hypermethylation).

**Table 2 ijms-21-00036-t002:** The epigenetic changes found in patients with T1DM: Histone modification.

Publication	Specific Target	Type of Cells	Results
(1st Author + Year)			
Miao, 2008 [46]	genome-wide + H3K9me2	monocytes	no differences
		lymphocytes	↑ H3K9me2
Miao, 2012 [47]	T1DM susceptible loci	monocytes	differences in H3K9Ac:
	+ H3K9Ac, H4K16Ac,		↑ *HLA-DQB1*, ↓ *HLA-DRB1*
	H3K4, H3K9, H3K27 me3	lymphocytes	no differences
Chen, 2009 [50]	genome-wide H4 acetylation	monocytes	↑ H4 acetylation
Liu, 2015 [51]	genome-wide H3 acetylation	CD4+ T cells	↓ H3 acetylation

H3K9me2—dimethylation of lysine 9 of histone 3. H3K9Ac, H4K16Ac—acetylation of lysine 9 of histone 3, and of lysine 16 of histone 4. H3K4, H3K9, H3K27 me3—trimethylation of lysine 4, 9, and 27 of histone 3. ↓ Decrease of histone modification. ↑ Increase of histone modification.

**Table 3 ijms-21-00036-t003:** The epigenetic changes found in patients with T1DM: RNA interference.

Publication	Specific Target	Type of Cells	Results
(1st Author + Year)			
Hezova, 2010 [54]	genome-wide	Tregs	↑ miR-510
			↓ miR-342 + miR-191
de Jong, 2016 [55]	genome-target	CD8+ T cells	↑ miR-98, miR-23b, miR-590
Sebastiani, 2011 [56]	miR-326	lymphocytes	↑ miR-326
Salas-Pérez, 2013 [57]	miR-21a + miR-93	mononuclear cells	↓ miR-21a + miR-93
Yang, 2015 [59]	genome-wide	mononuclear cells	↓ miR-146
			differences in 26 miRNAs

↓ Decrease of miRNA expression. ↑ Increase of miRNA expression.

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
