# Peer review of "Epigenetic Regulation in Etiology of Type 1 Diabetes Mellitus"

_ijms, 2019, doi:10.3390/ijms21010036_

Round 1
Reviewer 1 Report
The review is interesting and well-written. The author analyzes thoroughly the studies about the epigenetic mechanisms involved in the pathogenesis of DM type 1, as well as the epigenetic mechanisms per se. The manuscript deserves publication, but because the provided data are too many; visualization would make the manuscript easier for the reader.
Thus, besides providing figure 1, I suggest including extra figures and tables in the manuscript. More precisely:
Figures to depict the epigenetic mechanisms (DNA methylation, histone acetylation and methylation, and miRNA). Tables to describe the epigenetic changes found in patients with DN type 1. The type of epigenetic change, the specific target of the epigenetic change, and the type of cell in which the epigenetic change was detected.Author Response
First, I would like to thank you very much for all reviewer’ comments that were very useful and helped me significantly to improve the publication. A have accepted all of them.
You asked me to add more figures and tables. So I did according to their recommendation. I have added 3 figures and 3 tables.Las but not least, I have changed abbreviation “T1D” into T1DM”, and I have done some small technical corrections (in Figure 1, abbreviation “FOXP3” in cursive letter, correction of small interfering RNA - siRNA).
Thank you again for your recognition of our research
Marie Cerna
Reviewer 2 Report
The manuscript written by Marie Cerna represents a very detailed analysis of epigenetic aspects of the etiology of type 1 diabetes mellitus (T1DM). The author meticulously presented various possible mechanisms underlying the epigenetic regulation of this autoimmune disease.
Unfortunately, the article contains only one figure, which presents very general genetic complexities described in the text. Is it possible to enrich this article with additional figures / tables that will be more reader friendly and easier to remember?
Could the author add a paragraph presented a slightly broader description of the impact of the environment (i.e. nutrition, stress, chemicals and microbes) on epigenetic regulations of predisposing genes to T1DM?
Author Response
First, I would like to thank you very much for all reviewer’ comments that were very useful and helped me significantly to improve the publication. A have accepted all of them.
You asked me to add more figures and tables. So I did according to their recommendation. I have added 3 figures and 3 tables. You asked me to add a paragraph presented a slightly broader description of the impact of the environment on etiology of T1DM. So I did, see the paragraph 1.2. Environmental risk factors in T1DM.
Las but not least, I have changed abbreviation “T1D” into T1DM”, and I have done some small technical corrections (in Figure 1, abbreviation “FOXP3” in cursive letter, correction of small interfering RNA - siRNA).
Thank you again for your recognition of our research
Marie Cerna
Round 2
Reviewer 1 Report
The author addressed all the raised issues adequately.